# Empowering and Disempowering Motivational Climates, Mediating Psychological Processes, and Future Intentions of Sport Participation

**DOI:** 10.3390/ijerph19020896

**Published:** 2022-01-14

**Authors:** Nallely Castillo-Jiménez, Jeanette M. López-Walle, Inés Tomás, José Tristán, Joan L. Duda, Isabel Balaguer

**Affiliations:** 1Facultad de Organización Deportiva, Universidad Autónoma de Nuevo León (México) (UANL), Av. Universidad s/n, Cd. Universitaria, San Nicolás de los Garza 66455, Nuevo León, Mexico; jeanette.lopezwl@uanl.edu.mx (J.M.L.-W.); jose.tristanrr@uanl.edu.mx (J.T.); 2Faculty of Psychology, University of Valencia, Av. Blasco Ibañez, 21, 46010 Valencia, Spain; ines.tomas@uv.es (I.T.); isabel.balaguer@uv.es (I.B.); 3School of Sport, Exercise and Rehabilitation Sciences, University of Birmingham, Birmingham B15 2TT, UK; j.l.duda@bham.ac.uk

**Keywords:** motivational climate, basic psychological needs, soccer, self-determined motivation

## Abstract

Based on the conceptual model of multidimensional and hierarchical motivational climate the objective of this study was to test two models. One model (M1) of total mediation, testing the mediating mechanisms that explain why the motivational climate affects intention of continuity or dropout. Specifically, we test the mediating role of satisfaction/frustration of basic psychological needs and self-determined motivation, in the relationship between the players’ perception of the empowering and disempowering climate created by the coach, and the intention of young soccer players to continue/dropout the sport practice. The second model (M2) of partial mediation, contributes to knowing the mechanisms that link the antecedent variables included in the model (perceived empowering and disempowering motivational climate) and the outcomes (intention of continuity or dropout in sport). A total of 381 young male soccer players between 12 and 14 years of age (*M* = 12.41, *SD* = 0.89), completed a questionnaire package tapping into the variables of interest: players’ perception of the motivational climate created by the coach (empowering and disempowering), satisfaction/thwarting of basic psychological needs, self-determined motivation and the intention to continue/dropout sports participation. The hypothesized model was tested using a structural equation model technique with latent variables. The results of the partial mediation model were satisfactory (χ^2^= 120.92; *df* = 68; RMSEA = 0.045; CFI = 0.968; TLI = 0.957) and showed that need satisfaction and self-determined motivation partially mediated the relationship between the perception of the empowering climate and the intention to continue. Moreover, need satisfaction showed a positive and significant relationship with the intention to continue sports participation. Additionally, need thwarting and self-determined motivation totally mediated the relationship between the perception of the disempowering climate and the intention to dropout. Furthermore, needs thwarting was positively and significantly related to the intention to dropout of sports participation. Findings point to the importance of fostering empowering climates and preventing the creation of disempowering climates in the grassroots football.

## 1. Introduction

Research has consistently shown that motivational climates created by coaches are related to the quality of motivation of young athletes and with their enjoyment in sport activities [1,2,3,4,5,6]. The motivational climate in sports is referred to the psychological atmosphere and indicates what the coach does and says, and how he/she structures the environment in training and competitions [7].

This study uses a hierarchical, multidimensional model of the motivational climate [1] that is based on two contemporary motivational theories: The Achievement Goal Theory (AGT) [8,9] and the Self-Determination Theory (SDT) [10]. These theories applied to the setting of sports defend that the motivational climate created by coaches’ favors or hampers the quality of sport participation and optimum functioning of the athletes [4,11,12].

AGT states that the coaches’ motivational climate influences how athletes judge their competence and define their success [4]. In the motivational climate in sport, the existence of two dimensions is observed: the first refers to a task-involving motivational climate, in which the coach emphasizes cooperative learning between teammates, effort and skills improvement, and considers that all athletes have an important role in the team. The second dimension refers to the ego involving motivational climate, where the coach encourages rivalry, demonstrates unequal recognition of athletes based on their ability, and uses punishment for mistakes [13].

On the other hand, SDT [10] states that there are at least two interpersonal styles called autonomy support and controlling [10,14]. In an autonomy—supportive climate coaches favor athletes’ involvement in the decision-making process, recognize the athlete’s preferences and considers their perspectives, provides a rationale when requesting a specific task from the athletes, and offers them meaningful choices. In a controlling interpersonal style, coaches impose their point of view, pressure their athletes by imposing themselves in an authoritarian way, pay negative conditional attention, use rewards and controlling language, and tend to control the athlete’s personal lives [14]. The research framed in AGT [8,9] has reported that in the social contexts in which task involvement is promoted, adaptive cognitive, affective, and behavioral responses in athletes are favored [4,15,16]. On the other hand, research carried out based on SDT, in support of the theory, has proven that the social contexts that support autonomy favor intrinsic motivation while the controllers undermine it [17,18,19,20].

Duda [1] introduced and combined climate facets from AGT and SDT, proposing that coaches’ motivational climates can be more or less empowering and disempowering. Under these constructs, considering the different facets previously mentioned, Duda [1] proposes a multidimensional and hierarchal motivational climate where the creation of an empowering climate is characterized by promoting task involvement, supporting autonomy, and offering social support; while in the creation of a disempowering climate the coach favors ego involvement and shows a controlling style. Duda [1] defends that the climate is more or less empowering and more or less disempowering, and that a positive and adaptive experience in sport participation is more evident when it occurs in an empowering environment since, in this environment, task goal orientations, the satisfaction of psychological needs, more autonomous motivation, well-being, optimal functioning of the athlete, and intention to continue sports practice is favored [21]. However, when sports participation occurs in a disempowering climate, a negative and maladaptive experience is more evident because these controlling interpersonal styles hinder the optimal functioning of athletes, promoting ego goal orientations, need thwarting, controlled motivation, ill-being, and the desire to drop out [22].

To describe the way through which an empowering or disempowering motivational climate could influence athlete motivation and their responses, Duda’s [1] model introduces the role of the three psychological needs from SDT: competence (athletes perceive that they can meet the demands of the activity), autonomy (athletes feel they have a voice and choice when it comes to their sport participation), and relatedness (athletes experience a positive relationship with the teammates/and or the coach, and feel they belong and are cared for in that context). In the case that these needs are satisfied, is more probable that young athletes feel autonomously motivated. Autonomous reasons are considered beneficial motives for actively playing and continuing with sport. On the other hand, when the needs for competence, autonomy, and relatedness are actively frustrated by the coach, young athletes with high probability will feel that they engage in sport for controlled reasons. The model makes differential predictions in terms of circumstances in which young athletes will have intentions to continue or to drop out of sport. We could say that a positive and adaptive experience to youth sport will be produced when children participate in an empowering climate, and that their experience will be more maladaptive when young athletes participate in a disempowering climate.

This study intends to identify the mediating mechanisms in the relationship between the players’ perception of the empowering and disempowering climate created by the coach, and the intention of young soccer players to continue/drop out of the sport practice. Concretely, we tested the mediator role of satisfaction/frustration of basic psychological needs and self-determined motivation in the aforementioned relationships. To date, we do not know of any other study that includes all the variables of Duda’s [1] model that we introduce in this work. Therefore, this research empirically contributes to testing the theoretical propositions formulated in this model [1,21] to analyze if it provides validity. This work also proposes to offer sports coaches keys to understanding how a climate they create in their sports teams (empowering/disempowering) can favor the future intention to continue participating in the sport or increase the intention to drop out of sports participation, as well as the psychological mechanisms that contribute to this intention.

To date, research testing some parts of the model has been developed. For example, positive relationships have been found between the perceptions of a coach-created empowering climate and psychological need satisfaction [23,24,25] and between disempowering climate and psychological need thwarting [24,26,27]. It has also been reported that psychological need satisfaction is positively associated with autonomous motivation [17,24,28,29,30,31,32,33,34,35,36]. In contrast, psychological need thwarting was positively related to controlled motivation [20,24,26,31,37]. On the other hand, autonomous motivation has been positively associated with the intention to continue participating in the sport [26,28,33,38] while controlled motivation have been positively associated with the intention to dropout sport [24,39,40,41].

Based on Duda’s [1] assumptions and the evidence from previous research, our study focuses on analyzing the role that the perceived motivational climate (empowering and disempowering) created by the sports coach plays in the intention of continuing and dropping out of sports participation in young soccer players. Specifically, we will analyze in two models the mediating mechanism that the satisfaction and thwarting of basic psychological needs and self-determined motivation play in that relationship.

Specifically, we propose two objectives. The first is testing a model (M1) that hypothesizes that the perception of empowering climate is positively related to the psychological needs’ satisfaction, which in turn are positively related to self-determined motivation. On the other hand, the perception of disempowering climate is positively related to psychological needs’ thwarting, which in turn are negatively related to self-determined motivation. Self-determined motivation is positively related to the intention to continue participating in sports, and negatively related to dropping out sports participation (See Figure 1). Based on this objective, the following hypotheses are formulated to test mediation:

**Hypothesis** **1.**
*Psychological needs’ satisfaction and self-determined motivation will mediate the relationship between the perception of an empowering climate and the intention to continue sport participation.*


**Hypothesis** **2.**
*Psychological needs’ thwarting and self-determined motivation will mediate the relationship between the perception of a disempowering climate and the intention to drop out of sport participation.*


**Hypothesis** **3.**
*Self-determined motivation will mediate the relationship between psychological needs’ satisfaction and the intention to continue sport participation.*


**Hypothesis** **4.**
*Self-determined motivation will mediate the relationship between psychological needs’ thwarting and the intention to drop out of sport participation.*


The second objective is to deepen the study of mediation mechanisms; therefore, an alternative model of partial mediation (M2) is proposed that adds to the initial model (M1), the relationship between the perception of empowering/disempowering climates and the intention of continuing/dropping out, respectively, as well as the direct relationship between the satisfaction/thwarting of basic psychological needs and the intention to continue/drop out, respectively.

## 2. Materials and Methods

### 2.1. Participants

The sample consisted of 381 young male soccer players, all of them men from Nuevo Leon (Mexico) with an age range of 12 to 14 years (M_age_ = 12.41; *SD* = 0.89) with two hours of daily training (*SD* = 1.9) and two years with their current coach (*SD* = 1.37). Participants played at the national level and belonged to 37 different soccer teams.

Regarding power analysis, with this sample we should have enough statistical power to detect relevant relationships. According to sample size and statistical power calculations in multiple regressions, assuming a low effect size (f2 = 0.05) for a maximum number of predictors (5) and an alpha level of 0.05, in order to attain a statistical power level of 0.80, the required sample size would be 263 [42]. The study sample was composed of 381 male soccer players, thus was larger than the required to attain an adequate power level.

### 2.2. Instruments

To assess the participants’ perception of the motivational climate created by the coach, the Empowering and Disempowering Motivational Climate Questionnaire-Coach, EDMCQ-C [43] adapted to the Mexican context [23] was used. The following statement preceded the 34 items of the questionnaire: “Think about how things have gone in your team most of the time during the last 3 or 4 weeks,” and then the players evaluated, on the one hand, their perception of the empowering climate created by the coach (17 items) in the following three dimensions: Task-involving (9 items, e.g., “My coach encouraged players to try new skills”), autonomy-supportive (5 items, e.g., “My coach has given players different alternatives and options”) and social-supportive (3 items, e.g., “Whatever happens, we always have the coach’s support”). The perception of disempowering created by the coach (17 items) was evaluated in the following two dimensions: Ego-involving (7 items, e.g., “My coach substitutes players when they make mistakes”) and controlling coaching (10 items, e.g., “My coach is less friendly with players if they do not see things his/her way”). The items are answered on a Likert-type scale of five points that vary from strongly disagree (1) to strongly agree (5). Previous research has supported the internal consistency of the empowering and disempowering dimensions [43].

To assess the satisfaction of basic psychological needs (autonomy, competence, and relatedness), different scales were used that are described below. Satisfaction of autonomy was assessed with the Perceived Autonomy in Sport Scale, PASS [44], which consists of 10 items (e.g., “In my sport … I can give my opinion”). Responses are collected on a seven-point Likert-type scale that varies from not at all true (1) to very true (7). Satisfaction of competence was measured with the Perceived Competence scale of the Intrinsic Motivation Inventory, IMI [45], which consists of five items (e.g., “I am quite good at my sport”). Responses are collected on a seven-point Likert-type scale ranging from strongly disagree (1) to strongly agree (7). Satisfaction of relatedness was assessed with the Acceptance subscale of the Need for Relatedness Scale, NRS [46], composed of five items (e.g., “When I participate in my sport, I feel...supported”). Responses are collected on a five-point Likert-type scale that varies from strongly disagree (1) to strongly agree (5).

The mean of the three needs was calculated to assess the satisfaction of basic psychological needs. Previous research has confirmed the reliability of the three scales with Mexican samples [47].

Psychological need thwarting was measured using the Mexican version [48] of the Psychological Need Thwarting Scale [49]. The scale consists of 12 items divided into three four-item subscales assessing the perceived thwarting of personal feelings of autonomy, competence, and relatedness in the sport setting. The items are preceded by the phrase “In my sport…” Examples of the items for each subscale are, “I feel prevented from making choices about the way I train in my sport” (autonomy); “There are situations in my sport that make me feel incompetent” (competence); “In my sport, I feel that others don’t take me into account.” (relatedness). The instrument is answered on a Likert-type scale ranging from (1) strongly disagree to (7) strongly agree. Previous research has confirmed the instrument’s reliability in Mexican samples [31,50,51].

Self-determined motivation was assessed using the Spanish version of the Behavioral Regulation in Sport Questionnaire, BRSQ-6 [52], adapted for young soccer players [53], and used in the Mexican context [23]. The scale consists of 20 items divided into five subscales: intrinsic motivation, identified regulation, introjected regulation, external regulation, and no motivation; each subscale is measured with five items. The scale begins with the phrase: “I play soccer with this team…” The following are examples of each of the subscales: intrinsic motivation, “Because I enjoy it”; identified regulation, “because the benefits of soccer are important to me”; introjected regulation “because I would feel guilty if I left”; external regulation “because others push me to do it”; and no motivation “I still wonder why I continue.” Responses are collected on a Likert-type scale ranging from strongly disagree (1) to strongly agree (5). The reliability of this instrument has received support from previous research [53]. Following what is suggested by SDT, and consistent with previous studies [28,35,54], and specifically using the BRSQ [23] in this work, a self-determination index (SDI) was used, calculating the weight of each type of motivation according to its position on the self-determination continuum. Intrinsic motivation has the highest weight (+2); identified regulation a lower weight (+1); external regulation a negative weight (−1); and no motivation the most negative weight (−2). Introjected regulation represents the middle point of the self-determination continuum, and therefore it is not considered in the calculation of the self-determination index. High values in this index indicate high self-determined motivation.

To assess the intention to continue and the intention to dropout participation in soccer in the next season, five items used by Sarrazin et al. [40] were adapted to soccer and used in the current research. The players were asked to indicate if they agreed or disagreed according to what they thought at the moment they answered the five items of the instrument. Three of the items refer to the intention to continue (e.g., “I plan to play soccer next season”), while the other two items refer to the intention to dropout (e.g., “I plan to dropout soccer when the season ends”). The players answered with a Likert-type scale that varied from 1 (strongly disagree) to 5 (strongly agree). The questionnaire has demonstrated adequate psychometric properties in other studies in Mexican context [26,27,41].

### 2.3. Procedure

The participants were informed of the aim of the study, about the nature of their voluntary involvement in the study, and about the absolute confidentiality of their answers and data management. In addition, respecting ethical protocols and considering that the participants were minors, the coaches were asked, on behalf of the parents as responsible persons, to sign the informed consent for participation in the study. All the players voluntarily accepted to participate in the study. The mean time to respond was 20 min. Approval and permission were obtained from Sinergia Deportiva and the academies. Likewise, assent for participation was requested from the players of each of the academies. Later, the research project and the objective of the study were explained. After authorization was obtained, each of the different academy coordinators in the metropolitan area of Monterrey were contacted. After consent was obtained from each of the coordinators from the academies, the application of the instruments started. The procedure for collecting the information was the same in all academies, with all the collaborators following the same standardized protocol. Before starting the training, players were asked to answer the package of psychological questionnaires, except in some academies where the information was collected after training. The type of sampling was nonprobabilistic.

### 2.4. Data Analysis

As a first step, the factorial structure of each of the scales was verified by confirmatory factor analysis (CFA) [55,56]. Concretely, five CFAs were run: (1) for Empowering and Disempowering Motivational Climate Questionnaire-Coach, a five-factor model was tested, identifying the different subscales within each dimension (empowering: task involving, autonomy support, social support; disempowering: ego involving, controlling style); (2) for psychological need satisfaction, a three-factor model was tested identifying the three scales that measure the different needs (competence, autonomy, and relatedness); (3) for psychological need thwarting, a three-factor model was tested identifying the three thwarting subscales (competence, autonomy, and relatedness); (4) for self-determined motivation, a five-factor model was tested identifying the five subscales of the Behavioral Regulation in Sport Questionnaire (intrinsic motivation, identified regulation, introjected regulation, external regulation, and no motivation); and (5) for the intention to continue and the intention to drop out of sport participation, a bifactorial model was tested identifying the items that belong to each of the two subdimensions. To assess the fit of the models, absolute, incremental, and parsimony goodness-of-fit indices were considered. The following absolute goodness-of-fit indexes were used: the chi-squared value (χ^2^) and the ratio between χ^2^ and degrees of freedom (*df*), χ^2^/*df*, considering the latter with values below five as adequate fit for the factorial model [57]; the root-mean-square error of approximation (RMSEA), for which values below 0.08 represent an adequate fit to data [58]. The following incremental goodness-of-fit indexes were used: the comparative fit index (CFI), with values that vary between 0 and 1, and values greater or equal to 0.90 indicating a satisfactory fit [58]; the Tucker–Lewis Index (TLI) [59] for which values equal to or greater than 0.90 are considered indicators of satisfactory fit. Finally, the Akaike Information Criterion (AIC) [58] was used as parsimony goodness-of-fit index.

Descriptive statistics (range, mean, and standard deviation), reliability (Cronbach’s alpha and omega), and bivariate correlations between the study variables were estimated.

The hypothesized model was tested using a structural equation model technique with latent variables, using the corresponding subscales as indicators. The criteria used to determine the fit of the models were similar to those explained before for the CFA. For comparison of alternative models, the criteria for the incremental, absolute, and parsimony fit indices were used. For comparison of incremental fit indices, the difference between CFI and TLI was used. Previous studies suggest that a difference of 0.01 or less between the CFI values (ΔCFI < 0.01) or between the TLI values (ΔTLI < 0.01) of alternative nested models indicate practical irrelevant differences between the compared models [60,61]. For comparison of the absolute indices, the difference of the RMSEA was used. Chen [62] suggested that RMSEA differences of <0.015 between alternative nested models indicate irrelevant differences; therefore, the more parsimonious model should be selected. Finally, to compare goodness-of-fit indices of parsimony, the AIC was used; for this, values near 0 indicate a better fit of the model [58]. 

The bootstrap confidence interval (CI) as implemented in Mplus [63] was used to estimate the four indirect effects (IE) that are hypothesized in the model. This method allows for obtaining an estimate of the indirect effect (as the product of the regression coefficients involved in the mediation chain), and a 95% confidence interval (95% CI) for this indirect effect. When the confidence interval does not include zero, empirical evidence is obtained in favor of the indirect effect.

IBM SPSS statistical software was used to perform descriptive, correlational, and reliability analyses, MASTER v 24 was used to estimate confirmatory factor analyses, and Mplus 8.2 [64] was used for structural equation models.

## 3. Results

### 3.1. Confirmatory Factor Analysis

The first step was to analyze the factorial validity of each of the instruments using confirmatory factor analysis (Table 1). The factor loadings obtained in all the scales were significant (*p* < 0.05), ranging from 0.23 to 0.91; also, all the measurement instruments showed adequate goodness-of-fit indices.

### 3.2. Descriptive Analysis, Reliability, and Correlations

A seen in Table 2, the majority of the scales had satisfactory reliability (Cronbach’s alpha > 0.70), except the intention-to-continue scale (Cronbach’s alpha = 0.64; omega = 0.66). The correlation values were as expected (Figure 1); in other words, the perception of the empowering climate was positively and significantly associated with satisfaction of psychological needs (*r* = 0.40, *p* < 0.01), and these were positively with self-determined motivation (*r* = 0.17, *p* < 0.01). Likewise, the perception of the *disempowering* climate was positively and significantly associated with thwarting of basic psychological needs (*r* = 0.42, *p* < 0.01). These were also negatively and significantly associated with self-determined motivation (*r* = −0.50, *p* < 0.01). Finally, self-determined motivation was positively and significantly associated with the intention to continue (*r* = 0.26, *p* < 0.01) and negatively and significantly with the intention to drop out of sport participation (*r* = −0.39, *p* < 0.01). Correlations showed small–medium effect size [65].

### 3.3. Structural Equation Model and Indirect Effects

The hypothesized model (M1) offered satisfactory goodness-of-fit indices (χ^2^ = 180.16; *df* = 72, *p* < 0.01; χ^2^/*df* = 2.50; RMSEA = 0.063; CFI = 0.934; TLI = 0.917; AIC = 1361.9) and the regression coefficients obtained supported the hypothesized relationships between the variables (see Figure 2). Concretely, the results of the tested hypothesized M1 model indicate that the perception of the empowering climate created by the coach presents a positive and significant relationship with psychological need satisfaction (β = 0.50, *B_1_* = 0.60, *p* < 0.01). On the other hand, the perception of the disempowering climate has a positive and significant relationship with psychological need thwarting *(*β = 0.52, *B_4_* = 1.38 *p* <0.01). Psychological need satisfaction presents a positive relationship with self-determined motivation (β = 0.21, *B_2_* = 1.38, *p* <.01), while psychological need thwarting is negatively related with self-determined motivation (β = −0.53, *B_5_* = −1.83, *p* <.01). Finally, self-determined motivation is positively associated with the intention to continue (β = 0.26, *B_3_* = 0.05, *p* < 0.01) and negatively with the intention to drop out (β = −0.39, *B_6_* = −0.10, *p* < 0.01).

When testing the four indirect effects (IE) included in the model, the results showed that none of the confidence intervals included a zero value. Therefore, we confirmed that satisfaction of basic psychological needs and self-determined motivation mediated the relationship between the perception of the *empowering* climate and the intention to continue (IE = B_1_ * B_2_ * B_3_ = 0.04, 95% CI = [0.01, 0.09]). Likewise, self-determined motivation mediated the relationship between psychological need satisfaction and the intention to continue (IE = B_2_ * B_3_ = 0.07, 95% CI = [0.02, 0.14]) On the other hand, thwarting and self-determined motivation mediated the relationship between the *disempowering* climate and the intention to drop out (IE = B_4_ * B_5_ * B_6_ = 0.25, 95% CI = [0.15, 0.37]). Likewise, self-determined motivation mediated the relationship between psychological need thwarting and the intention to drop out (IE = B_5_ * B_6_ = 0.18, 95% CI = [0.12, 0.25]).

Finally, to deepen the study of mediation mechanisms, a contrast of two models was considered: the hypothesized model (M1) of total mediation and a partial mediation model (M2). The partial mediation model (M2) showed a satisfactory goodness of fit to the data (χ^2^ = 120.92; *df* = 68; χ^2^/*df* = 1.78; RMSEA = 0.045; CFI = 0.968; TLI = 0.957; AIC = 1355.54). Additionally, it was confirmed that the perception of the empowering climate directly and positively predicted the intention to continue (β = 0.20, *B_7_* = 0.30, *p* < 0.05); however, the perception of the disempowering climate did not have a direct effect on the intention to drop out (β = 0.10, B_8_ = 0.24, *p* > 0.05). Likewise, it was confirmed that psychological need satisfaction directly and positively predicts the intention to continue (β = 0.22, B_9_ = 0.27, *p* < 0.01), and psychological need thwarting has a direct positive effect on the intention to drop out of sports (β = 0.23, *B*_10_ = 0.21, *p* < 0.01). Finally, if we compare the fit of the two models (M1 and M2), we can conclude that there are relevant differences in fit between both models (ΔRMSEA = 0.018; ΔCFI = 0.034; ΔTLI = 0.040; AIC_M2_ < AIC_M1_), concluding that M2 presents a better fit compared to M1; thus, we could say that there are partial mediations for three of the tested indirect effects and a total mediation between the perception of the disempowering climate and the intention to drop out through psychological need thwarting and self-determined motivation. According to the squared multiple correlation coefficients, results indicated that the model explained 16% of variance for psychological need satisfaction, 18% for psychological need thwarting, 28% for self-determined motivation, 17% for intention to continue, and 20% of variance for intention to drop out of sport.

## 4. Discussion

Within the framework of AGT [8,9] SDT [10] and Duda’s [1] model of the motivational climate, the objective of this study, carried out with a sample of young football players, was to analyze the relationship between the players’ perception of the coach-created motivational climate (empowering and disempowering) and the player’s intentions to continue or drop out of sports participation, as well as the mediating role of psychological need satisfaction/thwarting and self-determined motivation in that relationship.

Based on Duda’s [1] conceptualization, first we tested a motivational model in which the perception of the empowering climate would promote the players psychological need satisfaction, while the perception of the disempowering climate would predict their psychological need thwarting. Need satisfaction would positively predict self-determined motivation while need thwarting would act as a negative predictor. Finally, self-determined motivation was postulated to have positive implications for the intention to continue participating in the future and negative implications for the intention to drop out of sports practice. Second, we deepened the analysis of the hypothesized mediating role of basic psychological needs (satisfaction and thwarting) and self-determined motivation in the model.

According to the SDT and Duda’s [1] model, the results of the structural equations model (M1) indicated that the young players’ perception of the empowering climate created by their coaches was positively related with player’s satisfaction of their psychological needs. These results are similar to those of previous studies conducted in the dance and sport contexts [24,25], where the importance of the empowering climate created by the coach on the athletes’ and dancer’s satisfaction of their psychological needs was shown. Additionally, in tune with the theory and in line with previous research, psychological need satisfaction was positively and significantly associated with self-determined motivation [28,33,34,36,40,66]. We also found, in accordance with the theory and with previous results in the literature, that self-determined motivation positively predicts the intention to continue sports participation [33,40]. On the other hand, in support of the theory and in line with previous research, we found that the players’ perception of the *disempowering* climate created by their coach is positively related with psychological need thwarting [24,25,27], and that these, in turn, act as a negative predictor of autonomous motivation [20,24,31,37], and as in previous studies, the latter is negatively associated with sports dropout [24,33,39,40,41].

The results of this model (M1) have allowed us to test the psychological mechanisms that explain why the motivational climate can affect sports continuity or dropout. Regarding the result of the second objective, in the M2, we have delved into the characteristics of mediation (total or partial).

We have found empirical evidence that supports that the psychological need satisfaction and self-determined motivation partially mediate the relationship between the empowering climate and the intention to continue (Hypothesis 1). This study also shows that self-determined motivation partially mediates the relationship between psychological need satisfaction and the intention to continue (Hypothesis 3). This has important implications for coaches, indicating the intervening mechanisms to achieve a future intention of staying in sports practice. Both the empowering climate and satisfaction of basic psychological needs present direct effects on the intention of future participation, and the mediating mechanisms in these relationships are established. The creation of an empowering climate facilitates the satisfaction of basic psychological needs of athletes, making them feel a like they have voice and a vote (autonomy), making them feel oriented toward mastery of the task and personal improvement (competence), and making them feel respected and connected (relatedness). Satisfaction of these basic psychological needs facilitates that the motives and reasons they practice sport are self-determined. All of this finally leads to young soccer players having future intentions to continue practicing their sport.

In contrast, we have found empirical evidence that supports that psychological need thwarting and self-determined motivation completely mediates the relationship between the perception of the disempowering climate and the intention to drop out of sports (Hypothesis 2). That is, the way that coaches favor that their players have the intention to abandon sport is through the thwarting of their need of competence, autonomy and relatedness, and impeding their self-determine motivation.

This study also shows that self-determined motivation partially mediates the relationship between psychological need thwarting and the intention to drop out of sports (Hypothesis 4). These results lead us to interpret that the perception of the disempowering motivational climate created by the coach has significant consequences on the future intentions to drop out of sports participation through thwarting of basic psychological needs and the athletes’ quality of motivation. The perception of a disempowering climate facilitates thwarting of basic psychological needs. When young players perceive that they are prevented from having a say in their sport (autonomy thwarting), that their coach makes them feel incompetent (competence thwarting), and when they feel rejected by those around them (relatedness thwarting), these feelings of thwarting are associated with low levels of self-determined motivation, which leads to an increase in their intention to drop out of sports participation. Thus, we can interpret that when coaches create disempowering climates, they favor the negative experiences that young soccer players have when participating in sport, as well as their intentions to drop out, indicating that thwarting of basic psychological needs and low self-determined motivation are mediating mechanisms in this relationship. In addition, thwarting of basic psychological needs has a direct negative effect on the intention to drop out of sports practice.

The players’ perception of the motivational climates created by their coaches are definitely key elements to achieve satisfaction or thwarting of young athletes’ needs, greater or less autonomous motivation, and their desire to continue or drop out of sports participation.

Specifically, the results demonstrate the importance of creating empowering climates and avoiding disempowering climates to achieve quality of participation in young players. This study provides empirical evidence that coaches have the possibility of creating an empowering environment in soccer, so players can live satisfying experiences that will lead them to decide if they want to continue participating in soccer. Likewise, we have seen that there is also a dark side to youth soccer. When coaches create disempowering climates, they promote young soccer players to have negative experiences in their sport and to want to drop out of it. In this case, thwarting of basic psychological needs and low self-determined motivation are the mediating mechanisms in this relationship (*disempowering* climate–intention to dropout).

This study also has some limitations. First, an objective evaluation of the coaches’ behavior is not included nor of their perception of the climate they create in their teams; this would have helped to have a complete idea of the study phenomenon. Second, this study focused on the intention to continue and the intention to drop out of sports. It would be interesting for future studies to focus on behaviors and not only intentions, that is, to consider actual dropout rates. Third, the study was obtained by self-reporting, and the sample consisted only of male athletes. Therefore, it would be important in future research to use objective measures and include women in the sample. Fourth, the study is cross-sectional in a soccer season, so we suggest developing longitudinal studies to more precisely understand the possible implications that these models could have in the sports context. Fifth, the reliability of the “intention to continue” scale was only adequate, which leads us to aim for future studies with higher reliability that would replicate our results. Finally, it would be necessary in the future to expand the sample size, expand the age range, and use a multilevel methodology.

This work has theoretical and practical implications. First, it offers empirical support of a theoretical model of the perception of empowering and disempowering climates from the conceptualization initially proposed by Duda [1] and developed in recent years by Duda and her collaborators [21]. In this model, in which the AGT and the SDT are integrated, processes that lead to the intention to continue and drop out of sports are presented. The practical implications point to the benefits of developing intervention and training programs that favor the development of empowering climates by sports coaches, since, as has been shown, these favor the intention to continue practicing sports through the satisfaction of basic psychological needs and the development of self-determined motivation. On the other hand, it is essential to instruct coaches to avoid behaviors that favor a disempowering climate because of the adverse effects this can have on the basic psychological needs of their athletes, their self-determined motivation, and finally, their intention to continue practicing sports [22].

## 5. Conclusions

To date, few empirical studies have analyzed the model and complete sequence of variables that are tested in this work. This research, carried out in a sports context, suggests that coaches should generate empowering motivational climates and avoid promoting disempowering climates, since they are key people who can influence athletes’ intention to continue or drop out of sports practice.

## Figures and Tables

**Figure 1 ijerph-19-00896-f001:**
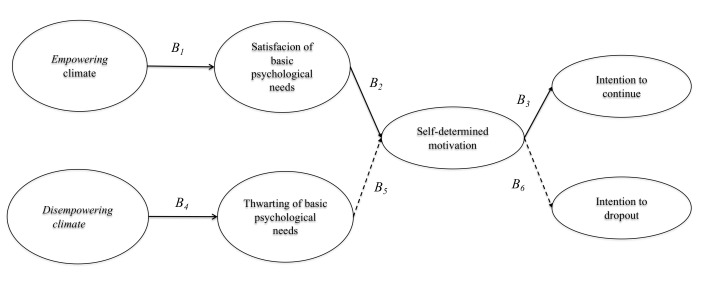
Graphic representation of the hypothetical model (M1). Note: Solid lines represent positive relationships and dashed lines negative relationships.

**Figure 2 ijerph-19-00896-f002:**
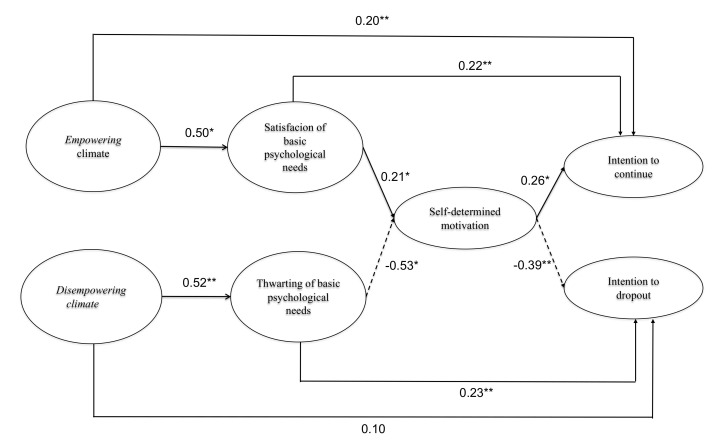
Structural Equations Model of the sequence Climate empowering and disempowering ® Satisfaction/Frustration of Basic Psychological Needs ® Self-Determined Motivation ® Future intentions to participate (M2). Note. This structural equation model predicts future intentions to participate from the perception of empowering and disempowering climates generated by the coach, with mediating effects of satisfaction/thwarting of basic psychological needs and self-determined motivation. Solid lines represent positive relationships; dashed lines represent negative relationships; bold lines (solid or dashed) represent significant indirect relationships. Standardized coefficient values are offered (β). * *p* < 0.05. ** *p* < 0.01.

**Table 1 ijerph-19-00896-t001:** Confirmatory factor analysis of the instruments used to measure the study variables.

Variables	Factor Loadings		Absolute Fit Indices				Incremental Fit Indices		Parsimony Fit Indices
	Min	Max	χ^2^	df	χ^2^/df	RMSEA	CFI	TLI	AIC
1. Empowering and disempowering climate	0.23	0.69	844.83	463	1.86	0.03	0.90	0.90	1038.83
2. Satisfaction of psychological needs	0.44	0.72	695.00	170	4.09	0.06	0.93	0.90	203.03
3. Thwarting of psychological needs	0.51	0.83	219.44	51	4.30	0.06	0.93	0.90	297.44
4. Self-determined motivation	0.51	0.77	348.39	142	2.45	0.04	0.92	0.90	482.39
5. Intention to continue and drop out	0.45	0.91	5.94	4	2.34	0.04	0.99	0.98	41.37

**Table 2 ijerph-19-00896-t002:** Descriptive statistics, reliability and correlations between variables.

Variables	Range	*M*	*SD*	1	2	3	4	5	6	7
1. Empowering Climate	1–5	4.23	0.55	(0.88/0.90)						
2. Disempowering Climate	1–5	2.42	0.68	−0.14 **	(0.82/0.85)					
3. Satisfaction of BPN	1–7	5.21	0.82	0.40 **	−0.08	(0.92/0.93)				
4. Thwarting of BPN	1–7	3.29	1.50	−0.03	0.42 **	0.01	(0.92/93)			
5. Self-determined motivation (SDI)	−3.75 12	5.41	4.16	0.31 **	−0.36 **	0.17 **	−0.50 **	--		
6. Intention to continue	1–5	4.42	0.81	0.34 **	0.08	0.34 **	−0.03	0.26 **	(0.64/0.66)	
7. Intention to dropout	1–5	1.65	1.08	−0.16 **	0.26 **	−0.16 **	0.37 **	−0.39 **	−0.30 **	(0.75/0.77)

Note. BPN = Basic psychological needs; SDI = Self-determination Index. The values in parentheses on the diagonal correspond to the reliability (Cronbach’s alpha/omega) of each scale. *** p* < 0.01.

## Data Availability

The data supporting the results reported in the study can be viewed at the following link: https://www.dropbox.com/sh/rnix2fyblugva7v/AABp14yolzx6KJBryrQK4PD8a?dl=0 (assessed on 21 November 2021).

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
