# Peer review of "Empowering and Disempowering Motivational Climates, Mediating Psychological Processes, and Future Intentions of Sport Participation"

_ijerph, 2022, doi:10.3390/ijerph19020896_

Round 1
Reviewer 1 Report
-The type of statistical analysis is not clear in the abstract and the specific results (goodness-of-fit indices) are not shown.
-The manuscript presents an adequate development of the theoretical foundations that support the models to be tested.
-The knowledge gap to be addressed is clearly stated.
-It is recommended to decrease the number of self-citations of the research team.
-The study contributes to generating empirical evidence of current and relevant theoretical models in the field of sport psychology.
-Clarify that the type of sampling was non-probabilistic.
-Add information on how the instruments used are scored and interpreted.
-Clarify whether written informed consent was obtained from each participant, in addition to academic authorization.
-The CFA for each instrument guarantees the internal validity of the study. Which shows evidence of methodological rigor.
-The use of tables and figures is justified, however the figures could improve their resolution or graphic quality.
Author Response
"Please see the attachment."

Reviewer 2 Report
I have no doubt that what you are presenting exists in the coaching/athlete world exists. An empowered motivational climate will influence future intention to participate. I have no doubt that disempowering motivational climate, and mediating psychological processes affect future intentions of sport participation. I believe all of that is a given.
What bothers me about this study is the choice of participants; the r squared factor of the correlations, and the meaningfulness of those numbers on the model developed.
- The age of the participants is troubling - I question whether a 12 - 14 year old with a M of 12.41 years really has the capacity to give thoughtful answers to the instruments used.
- The majority of the Cronbach alpha scores were satisfactory at .70 (but not great) and the intention to continue scale was .64. I find these scores telling which may be linked to the age of the participants and their ability to stay on task.
- Then there is the issue of r squared. As you know, meaningfulness as related to correlations depends on
The significance. which you found,
The r? Which you report,
The coefficient determination of the r squared? Which you did not discuss?
The meaningfulness of the r squared for your correlations? Which you did not report.
Considering that the highest r reported was a -.50, perhaps you need to discuss what these mean? As you know this means that only 25 percent of the variability is supported - or that 75 percent is not accounted for - what do you think is that .75? What factors are playing into that number?
Another factor that I find troubling is that the variability is not considered when developing the model - it would seem that the model would have questionable meaning if the data imputed is questionable?
Statistics are only statistics without meaning - and unfortunately your short paragraph on limitations needs to address the issues above.
Again, I believe in the truthfulness of what this study is trying to support, but take a little time to discuss the problems above.
And finally, stating that coaches should be instructed to avoid the behaviors that favor a disempowering climate- is too little. As you well know, much is going on and coaches need some serious coaching on coaching the worth of the individual and not the objective worth of playing the game. Maybe that is your point - perhaps a model would prove the point, but make sure your model has the power of what you discuss.
Last. Numerous errors in the references if you are following a specific reference management source. Thirteen errors, I believe.
Check:
Aquirre
Ames
Browne
Byrne
Duda 2016 and 2007
Lopez-Walle --- autonomy support
McAuley
Monteiro
Newton
Nicholls
Pulido
Ryan
Author Response
"Please see the attachment."

Reviewer 3 Report
General comment: The paper is well-conceived and theory-driven. Therefore, I think the results can make a nice contribution to the existing literature, while also having clear practical implications for coaches. Nevertheless, I have some concerns about the sample size (given the complex models tested) and the rationale for focusing on male players only, on a single sport, and in that specific age range. Consequently, I think there’s room for improving the introduction section accordingly, better specifying the reasons why the authors chose this particular sample, as detailed later on.
Title: the title sounds to me informative, but a bit long and not particularly catchy; maybe you could consider rephrasing it to make it more concise and intriguing.
Abstract: lines 15-20 contain a quite long and not so clear sentence; please consider rephrasing it to make it simpler and more to the point.
Introduction
As previously mentioned, I would better contextualize the theories presented with respect to youth players and specify whether previous studies were conducted on adults or adolescents, hence better showing the need to study such a young population. Additionally, I would make it clearer why you decided to test two models (and not just the complete one): is it something really needed? What are the advantages and disadvantages?
I would also introduce basic psychological needs when talking about SDT, since you then measure them in the study.
ll. 138-145: I would move this before hypotheses are formulated, to stress the importance of conducting such a study
I would consider adding a second figure to depict Model 2 more clearly
Materials and Methods
Participants
Did you conduct a priori power analysis to determine sample size, also given the complexity of your models? If not, could you do so retrospectively to provide the readers with achieved power for your analyses?
Also, at what level did participants play? How many teams were involved in the study? Please specify.
Was there a particular reason you only focused on male players?
Instruments
Were all of them adapted/have been used with such young players before? Please specify.
Also, could you please report the internal consistency of the instruments in your sample?
Please provide an example of an item for each scale assessing basic psychological needs.
Procedure
Were parents informed about the purpose of the study, since participants were under the age of 18? Please provide additional information on consent given the regulations of your country. Also, did you ask for approval from the Ethical Board of your home university?
Data analysis
For the mediation models, did you consider the variables as observed or latent? Please specify.
Results
Are complete results for CFAs available somewhere, maybe Supplementary materials?
Please uniform “-“ and “−“ in text and tables.
l. 363: What does “EI” stand for? Please specify. Also, I think it would help to use the same letters (e.g., B1, B2, etc.) in text and in the figure (same for the other letters mentioned in l. 369).
Discussion
ll. 403-404: Please specify how you deepened the analysis of the hypothesized mediating role of BPNs and self-determined motivation.
l. 411 and following: Please specify the populations considered in the studies you cite: were all these studies conducted on youth players?
l. 475: I would be more cautious and use “could” instead of “will”.
I would mention the possibility to consider actual dropout rates in future studies (i.e., focusing on behaviors and not only intentions).
Author Response
"Please see the attachment."

Reviewer 4 Report
The study focuses on the relationship between the players' perception of the empowering and disempowering climate created by the coach, and the intention of young soccer players to continue/drop out of the sports practice. The authors present the statistics in providing the response to the hypotheses. I suggest to expand conclusions.
The study is well organized and shaped and I suggest accepting it for publication.
Author Response
"Please see the attachment."
